# The Potential Roles of Prophages in the Pathogenicity of *Klebsiella pneumoniae* Strains from Kenya

**DOI:** 10.3390/antibiotics14111145

**Published:** 2025-11-12

**Authors:** Juliah K. Akhwale, Ivy J. Mutai, Janet Y. Nale

**Affiliations:** 1School of Biological Sciences, Jomo Kenyatta University of Agriculture and Technology, Nairobi P.O. Box 62000-00200, Kenya; jakhwale@jkuat.ac.ke; 2Becky Mayer Centre for Phage Research, Division of Microbiology and Infection, University of Leicester, Leicester LE1 7RH, UK; ijm24@leicester.ac.uk; 3Antimicrobial Resistance & Bacteriophage Biology Section, Kenya Institute of Primate Research, Karen, Nairobi P.O. Box 24481-00502, Kenya; 4Centre for Epidemiology and Planetary Health, School of Veterinary Medicine & Biosciences, Scotland’s Rural College, Inverness IV2 5NA, UK

**Keywords:** *Klebsiella pneumoniae*, prophages, bacteriophages, antimicrobial resistance, pathogenicity, virulence genes

## Abstract

**Background/Objectives:** Antimicrobial resistance (AMR) in *Klebsiella pneumoniae* poses a serious threat to healthcare, especially in sub-Saharan Africa (SSA). To complement AMR infection control in Kenya, here, clinical and environmental genomes were investigated to determine the potential roles prophages play in *K. pneumoniae* pathogenicity. **Methods:** Prophages were extracted from 89 Kenyan *K. pneumoniae* genomes. The intact prophages were examined for virulence genes carriage, and their phylogenetic relationships were established. **Results:** Eighty-eight (~99%) of the genomes encode at least a single prophage, and there is an average of four prophages and 2.8% contributory genomes per bacterial strain. From the 364 prophages identified, 250 (68.7%) were intact, while 58 (15.9%) and 57 (15.7%) were questionable and incomplete, respectively. Approximately, 30% of the intact prophages encode 38 virulence genes that are linked to iron uptake (8), regulation (6), adherence (5), secretion system (4), antiphagocytosis (4), autotransporter (4), immune modulation (3), invasion (2), toxin (1) and cell surface/capsule (1). Phylogenetic analyses revealed three distinct clades of the intact prophages irrespective of their hosts, sources and locations, which support the plasticity of the genomes and potential to mediate horizontal gene transfer. **Conclusions:** This study provides first evidence showing the diverse prophages that are encoded in *K. pneumoniae* from SSA with particular focus on Kenyan strains. This also shows the potential roles these prophages play in the pathogenicity and success of *K. pneumoniae* and could improve knowledge and complement control strategies in the region and across the globe. Further work is needed to show the expression of these genes through lysogenisation.

## 1. Introduction

*Klebsiella pneumoniae* is a human opportunistic biofilm-forming bacterium that causes pneumonia, bacteremia and surgical site and urinary tract infections globally [1,2,3]. *K. pneumoniae* is also considered a priority pathogen by the World Health Organisation, and several strains are resistant to frontline antibiotics including cephalosporins, aminoglycosides and carbapenems [4,5,6]. Importantly, the continual emergence, existence and spread of antimicrobial-resistant and hypervirulent *Klebsiella* strains is of “critical” concern [7,8]. Thus, the combined impacts of these strains and classical *K. pneumoniae* make disease management extremely difficult. Also, this can lead to considerably high morbidity and mortality rates in nosocomial as well as community-acquired infections and from environmental sources, especially contaminated water [7,9].

Besides the problem of antimicrobial resistance (AMR) as indicated above, *K. pneumoniae* possesses various virulence indicators such as siderophores, capsules, lipopolysaccharide and fimbriae that play diverse roles in the pathogenicity of the bacterium [1,10,11]. Despite improved hospital management, increased surveillance and interest in the development of alternative and better treatment strategies, *K. pneumoniae* infection remains worryingly high [12,13]. This is due to the grossly inadequate antibiotic options currently available and the dwindling innovations to target and control emerging antibiotic-resistant phenotypes, thus limiting the prospects of alleviating this persistent health threat [12]. To mitigate this deficiency and curtail the impending health crisis associated with *K. pneumoniae*, there is a clear need to identify key genetic factors that control, mediate and disseminate virulence genes among the strains [14]. This will help to ascertain the role these factors play in the diversity, pathogenicity and emergence of new phenotypes and enable the development of more targeted controls for the pathogen.

The impact of AMR is predicted to be worse in low- and medium-income countries (LMICs), particularly in regions within sub-Saharan Africa (SSA) by the estimated year 2050, due to poor antibiotic stewardship and other underlying endemic infections [15,16]. Pertinent to Kenya, *K. pneumoniae* infection is a leading cause of death in humans, and ~6200 fatalities from the infection were reported in the year 2019 alone [17]. The prevalence is particularly high in children due to behavioural influences and low immunity, and thus, ~92% positive *K. pneumoniae* recovery was recorded from children (under 16 years old) in Kenya; importantly, ~64% of the isolates were identified as multidrug-resistant (MDR) [6,18]. In recent years, global *K. pneumoniae* MDR sequence types (STs) have emerged and high-risk lineages such as ST14, ST15, ST17, ST147, ST307, and ST607 were reported from Kenya [5,19,20]. Although the AMR profiles and virulence factors of these strains were also reported in the region, it is still unclear what factors are responsible for the spread and success of these strain sub-types. Thus, this study seeks to disentangle the mobile genetic elements that carry, drive and potentially disseminate the virulence factors within *K. pneumoniae* genomes, with particular interest in isolates from Kenya.

Most bacteria encode abundant integrated bacteriophages (also known as phages—bacterial parasitic viruses) called prophages in their genomes [21]. Prophages can promote the diversity, pathogenicity, ecology and physiological processes of their bacterial hosts via horizontal gene transfer. This can cause the emergence and evolution of novel phenotypes that are responsible for infection outbreaks, as shown in many bacterial systems [21,22,23,24]. Pertinent to this study, previous reports have shown that genomes of clinical *K. pneumoniae* isolates from various geographical areas around the world encode diverse and multiple prophage elements [25,26]. These prophages were shown to contribute to the evolution of the pathogen and confer additional phenotypes [25,26]. Specifically, prophages were also reported to play an important role in the pathogenicity of *K. pneumoniae* by encoding for virulence genes such as those responsible for toxin expression, efflux pump regulation, and *pks* island, which is responsible for the production of bacterial secondary metabolite genotoxin [27,28].

Following up on these studies and to ascertain the potential role prophages play in the pathogenicity of *K. pneumoniae* isolates from Kenya, this study strategically targeted and examined a set of 89 publicly available genomes [19]. The genomes were generated from a previous study that characterised isolates from five counties in the country and included their associated metadata, such as AMR and virulence genes, and ST profiles [19]. The strains are rich genomic resources encompassing isolates from community-acquired and healthcare-associated infections and hospital environments, which were collected between the years 2015 and 2020 [19]. Here, the genomes were subjected to extensive bioinformatics analyses to extract the prophage elements encoded thereof and to characterise them on the basis of their abundance and completeness per bacterial strain using a previously published prophage classification system [23]. Further analyses focused on identifying the virulence genes encoded by the intact prophages, assigning the genes to functions, and linking them to *K. pneumoniae* pathogenicity. Finally, phylogenetic relationships of the intact prophages were established to demonstrate their relatedness and associations.

## 2. Results

This study examined the carriage of diverse prophages in 89 *K. pneumoniae* genomes from Kenya. The prophages were expressed by their number, completeness and contributory genome content in each *K. pneumoniae* strain examined [23,29]. The virulence genes from the intact prophage genomes were further determined and linked to the pathogenicity of *K. pneumoniae* as shown in previous studies [27,30].

### 2.1. K. pneumoniae Strains from Kenya Carry Multiple and Diverse Prophages in Their Genomes

From the analysis of the 89 *K. pneumoniae* genomes using PHASTEST, a total of 364 diverse prophages were extracted with an average carriage of 4.09 prophages per bacterial strain. Only one strain, KK021, lacked any prophage in its genome, but the rest of the strains were found to encode at least a single prophage in each genome (Figure 1A). The highest prophage carriage per strain was eight, and this was identified in the MRSN825006, KKP057 and MRSN570493 genomes (Figure 1A).

In addition to the prophage abundance, the comparable cumulative genome contributions by the prophages carried in each bacterial strain was also analysed and is shown in Figure 1A. The prophage genome contribution per prophage-encoding bacterial strain averaged 2.8% and ranged from 0.13% (from strain MRSN824906) to 6.82% (from KKP026 genome) of the 5.05–5.76 Mb bacterial host genomes. It was observed that the prophage carriage in the *K. pneumoniae* strains was not commensurate with the prophage abundance in each genome. For example, strain KKP026, which has the highest contributory prophage genome content, encodes six prophages, but other strains with equal or higher prophage numbers have much lower prophage genome contribution to their bacterial host (Figure 1A,B) [23,29]. This depends on the genome sizes of the prophages.

Expressing the diverse prophage contents according to their completeness using a previous prophage classification system [23,29] showed that 63 of the *K. pneumoniae* strains were ‘low’ prophage carriers with up to five prophages identified from each of the genomes. Only 23 of the strains were ‘moderate’ prophage carriers, with 6–7 prophages identified in their genomes. Only three strains fell under the ‘high’ prophage carrying category, with eight prophages found in the genomes of the bacterial strains in this group (Figure 2A). This diverse prophage carriage was unrelated to the sources or county locations where the isolates were identified (Figure 2A).

Based on the prophage completeness, 250 prophages (68.7%) identified were intact while considerably fewer, and 58 (15.9%) and 57 (15.7%) of the prophages were questionable and incomplete, respectively (Figure 2B). Obviously, the intact prophages dominated the genomes compared with the incomplete and questionable prophages cumulatively (Figure 2B).

Considering the frequency of occurrence (267) of each of the prophage types and abundance in the *K. pneumoniae* genomes, the majority (a total of 100 occurrences) lacked any intact (4), incomplete (47) and questionable (49) prophages. The prophage load of one is being represented by intact (15), incomplete (29) and questionable (26), which accounted for the cumulative 71 occurrences of each of the prophage types in the strains. The rest of the intact prophage loads (prophage load of two or more) are found mostly in the intact prophages than in the questionable and incomplete ones (Figure 2C).

### 2.2. K. pneumoniae Intact Prophages Encode Several Virulence Genes That Are Related to Pathogenicity

Following the establishment of the prophage diversity within the *K. pneumoniae* genomes, further analyses were conducted to determine the potential impact the prophages may have on the pathogenicity of their hosts. Thus, the genomes of the extracted intact prophages (due to their potential to infect their hosts and mediate horizontal gene transfer) were exclusively investigated for the carriage of virulence genes [23].

From the analyses of the extracted 250 intact prophages using Vfanalyzer in the Virulence Factor Database (VFDB), 38 different virulence genes were identified and classified into 10 virulence categories [31]. The categories are represented by the functions: iron uptake (eight), regulation (six), adherence (five), secretion system (four), antiphagocytosis (four), autotransporter (four), immune modulation (three), invasion (two), toxin (one) and cell surface/capsule (one) (Figure 3, Appendix A). From these genes, a total of 529 gene copies were collated and found to be highly variable within the prophage genomes. The most frequent gene was *sigA*/*rpoV* with 299 gene copies found in the genomes (Appendix A).

Most of the virulence genes, eight genes (~21% of the genes) and 56 gene copies (10.6% of the total gene copies) that were successfully identified from all the prophages are linked to iron uptake (Appendix A). The genes *ybtQ*, *ybtP* and *ybtX* are linked to yersiniabactin (Ybt), a siderophore-dependent iron uptake system that is encoded within the pathogenicity island of many pathogenic bacteria [32]. The *pvdL* gene encodes for the pyoverdine chromophore synthetase and, together with *hutD*, help in iron acquisition [33,34]. Both *hbpB* and *hbpA* genes facilitate hemin and iron utilisation under low-iron conditions [35]. The siderophore acinetobactin binds iron with high affinity, uses the BauA TonB-dependent outer membrane receptor and utilises proton-motive force and an inner membrane ATP-binding cassette (ABC) transporter composed of BauB, BauC, BauD and BauE to transport iron into the cell [36].

The second highest set of virulence factors are linked to regulation processes with six genes: *phoP*, *gacA*, *sigA*/*rpoV*, *rpoS*, *bvrR* and *regX3* (~16.7% of the genes) and 335 gene copies (63.3% of the total gene copies) from the intact prophages. These genes encode for regulatory proteins that modulate the expression of iron acquisition, thereby influencing bacterial virulence [37].

The five genes *papA*, *APH_0632*, *ECVR50*, *sfbx* and *tcpI* (~13.9% of the genes) and 22 gene copies (4.2% of the total gene copies) are linked to adherence, a critical step in bacterial pathogenesis that is required for colonisation of new hosts [38]. This process is aided by adhesins, cell surface factors or appendages such as fimbriae and pili, which facilitate attachment and colonisation [39,40,41]. Bacterial adhesins are implicit targets for bacterial infection prevention or treatment.

Four genes, *ospC4*, *fliR*, *fliY* and *ompD* (~11.1% of the genes), with 52 gene copies (9.8% of the total gene copies) are associated with the secretion system. The *ospC4* codes for Mxi-Spa TTSS effectors are controlled by *virB*, while *fliR* and *fliY* code for flagella cluster I. *ompD* is an outer membrane precursor involved in the adhesion and invasion of colonic epithelial cells as shown in human Caco-2, a colorectal adenocarcinoma cell line [42].

Four additional genes, *algZ*, *algW*, *algJ* and *mucD* (11.1% of genes identified), and a total of 31 gene copies (5.9% of total gene copies detected) are related to alginate regulation and linked to antiphagocytosis. This process enables bacteria to destroy or weaken the phagocytotic machinery cells, and it is characterised by the injection of effector proteins directly into the cytoplasm of the host cells [43]. These effector proteins can interfere with specific cellular targets, e.g., disruption of the cytoskeleton, NF-κB or MAPK pathways, blocking the production of cytokines or inducing apoptosis [44].

The four genes *cdiA*, *cdiB*, *ehaB* and *agn43* (11.1% of genes identified) and a total of 11 gene copies (2.1% of total gene copies detected) are autotransporter genes. These encode a superfamily of proteins in Gram-negative bacteria that act as substrates of the type V secretion pathway and are involved with adhesion, colonisation, cell mobility, biofilm formation and cytotoxicity [45].

The three genes *hasD*, *vlh* and *M3Q_285* (8% of genes identified) and a total of six gene copies (1% of total gene copies detected) are linked to immune modulation. This strategy helps bacteria evade killing by phagocytes. *Klebsiella* can evade the host immune response by producing capsules to avoid phagocytosis, surviving the intracellular environment of phagocytes and degrading antibodies; or through antigenic variation [46].

Other genes that occurred less frequently were *cheD* and *nleG6-3* (~6% and three gene copies), involved in invasion (bacterial entry into host eukaryotic cells) [47,48]. The toxin was represented by the *sypC* gene (one gene copy), which is required for syringopeptin biosynthesis [49]. Lastly, the cell surface component gene, *sugC* (one gene copy), is important for host cell adhesion and invasion [50].

### 2.3. K. pneumoniae Intact Prophages Are Genetically Diverse and Formed Three Distinct Lineages

Phylogenetic analysis of the 250 intact prophage genomes revealed their genetic diversity and relationships, and three distinct clades (Clades 1, 2 and 3) were identified (Figure 4). Clade 1 (grey) is the largest of the three clades and consists of 47.6% (119) of the genomes, and this is followed by Clade 2 (green), which encompasses 34% (85) of prophage genomes. The final clade, Clade 2 (pink), is the smallest with 18.4% (46) of the intact prophage genomes. The clades include variable prophages relating to their bacterial hosts that were isolated from the different sources (UTI, hospital environment and skin) and the five locations/counties (Kilifi, Nairobi, Kericho, Kisii and Kisumu). We found no occasion where a particular clade exclusively contained prophages from bacterial strains that were isolated from a single source or location. Therefore, the prophages were distributed widely across the various clades irrespective of their host sources. For example, the seven intact prophages extracted from the KKP001 strain were distributed across two clades, Clade 1 and 2, with prophages KKP001_1, KKP001_3, KKP001_6 and KKP007 and KKP001_2, KKP001_4 and KKP001_4 in each of the clades, respectively. Similarly, the six prophages from strain KKP034 were distributed within Clade 1 (KKP034_1, KKP034_3, KKP034_4, KKP034_5 and KKP034_6) and Clade 3 (KKP034_2). Furthermore, the five prophages of MRSN613318 were distributed within all three clades—Clade 1 (MRSN613318_1 and MRSN613318_5), Clade 2 (MRSN613318_4) and Clade 3 (MRSN613318_2 and MRSN613318_3). However, for the three prophages of strain KKP037, the prophages (KKP037_1, KKP037_2 and KKP037_3) were all found in one clade, Clade 1. Although the prophages extracted from a single bacterial host may be distributed in different clades, they were all associated with their host source/niche.

## 3. Discussion

The growing threat of bacterial infections in the human global health sector, especially in LMICs, calls for an urgent, effective and strategic One Health mitigation approach [51,52]. Genetic studies provide powerful revolutionary capabilities to unravel the salient genetic markers that can be deployed for effective control and monitoring of bacterial infections. This may include the identification, characterisation and development of targeted monoclonal antibodies for effective vaccine design, typing and surveillance tools and phage receptors for therapeutic and diagnostic purposes [53,54,55]. To complement these control strategies, genome analyses can also reveal vital information on the carriage of mobile genetic elements (such as prophages) in host genomes and their potential influence on the success of pathogenic bacteria [23]. Thus, here, there was a focus on *K. pneumoniae,* a leading cause of mortality and morbidity in humans, especially children in Kenya, and indeed, a leading cause in the wider SSA and globally [18,19]. Our analyses showed that the Kenyan genomes carry abundant prophages that encode various virulence genes that are associated with the pathogenicity of *K. pneumoniae*, as reported previously, and that are associated with many other enteric bacteria such as *S. enterica*, *Clostridioides difficile* and *Escherichia coli* [23,29,30,56,57].

Prophages are known to mediate genetic acquisition and exchange between bacteria or from gene fragments and plasmids from other pathogens or the environment via horizontal gene transfer, leading to the possibility of lysogenic conversion [25,26,30,58]. Prophage integration frequencies are high in *K. pneumoniae*, and all 254 previously examined strains from a single study showed prophage carriage in every single genome, which supports the plasticity of the genomes [30]. The prophages can integrate in specific metabolic pathways, tRNA and transporter genes [25,30]. Thus, our approach of exploring the prophages in *K. pneumoniae* and their potential roles in pathogenicity is an established phenomenon. However, to the best of our knowledge, this report is the first to show the prophage content of strains from Kenya, especially from genomes of strains found in both clinical and environmental settings from a recent infection surveillance study in five counties in the country. This is also the most recent information on the genetic impact of prophages on *K. pneumoniae* strains in SSA as whole.

Of the 89 *K. pneumoniae* genomes analysed here, our observation that a high proportion (99%, 88 of 89) carry at least a single prophage (whether intact, incomplete or questionable prophages) in their genomes concurs with previous report [25]. Also, our findings showing that diverse prophages are abundant in naturally occurring *K. pneumoniae* strains are also supported by previous studies in other parts of the globe [25,26,27,28,30]. Additionally, our observation that the intact prophages were more abundant in the genomes examined here compared with the incomplete and questionable ones cumulatively concurs with previous work conducted on *K. pneumoniae* clinical isolates, irrespective of country/geographical location [25,30,59]. This observation contrasts findings on other bacteria, however, such as *Salmonella*, in which a greater proportion of the prophages identified were rather incomplete [23]. Some of the discrepancies of prophage abundance in bacteria may be attributed to either the bioinformatics tools or thresholds of genetic markers used for identifying and extracting the prophages, as shown in *Salmonella* [23]. However, since the study on *Salmonella* used comparable strategies employed here and elsewhere on *K. pneumoniae* [59], our findings strongly support the phenomenon that prophage abundance is indeed different in bacterial species, at least between *K. pneumoniae* and *Salmonella* [23].

Our observation that a single strain lacked any prophage in its genome is novel and particularly of interest, and this is generally extremely rare in naturally occurring bacteria. However, the lack of prophages in this strain can be helpful in its selection as a propagation host for therapeutic phages against *K. pneumoniae* infection [60]. This would help to provide clonal phages and prevent contamination of the bacterial lysate, phage suspension or product with potentially induced viral fragments [60]. Prophage induction in *K. pneumoniae* is widely mediated using DNA-damaging agents such as mitomycin C and triclosan [61]. However, spontaneous prophage induction and release can still occur through effect of diets, antibiotics, bacterial metabolites, gastrointestinal transit, inflammatory environment, oxidative stress and quorum sensing in the gut to produce viral particles [62]. These induced phages and other contaminants such as genetic materials and components, including unassembled phage tail fibres, can cause resistance response in *K. pneumoniae* by inducing loss of capsule and conferring localised immune response [63,64]. In addition, these contaminants can be extremely difficult to remove or purify; thus, exploring strains that lack prophages could be helpful for therapeutic phage preparations [65].

Only very few *K. pneumoniae* strains were reported to completely lack intact prophages [59], and we have also observed this feature in four of the genomes here (in addition to the one lacking any prophage). When the intact prophages were characterised, only 30% (75 of the 250 intact prophages identified) were found to encode at least a single virulence gene, which relates to *K. pneumoniae* pathogenicity (Figure 3). This suggests that although these prophages are abundant in clinical as well as environmental strains, including the STs that were examined here, only a few of them carry genes that are related to *K. pneumoniae* virulence [27,30]. The virulence factors reported in these studies are related to *K. pneumoniae* toxin and efflux pump regulators, as well as the phage toxin–antitoxin modules reported in various prophages [27,30]. Specifically, genes related to invasion were detected in prophage ST512-KPC3phi13 (GenBank accession number QBQ71533.1) [66]. Similarly, the regulator gene (MarR-like) was also detected in prophages ST13-OXA48phi12.5, ST13-OXA48phi12.3, ST16-OXA48phi5.2, ST101-KPC2phi6.3, ST405-OXA48phi1.3, ST11-VIM1phi8.2 and ST15-VIM1phi2.1 (GenBank accession numbers QEA09493.1, QBP27467.1, QBP28293.1, QBQ71610.1, QBP28507.1 and QBP28244.1, respectively) [27]. The most abundant virulence gene, the phage-encoded sigma factor *sigA*/*rpoV* gene is abundant in many systems and acts as an auxiliary gene that can alter bacterial physiology, regulate the prophage’s life cycle and influence host–pathogen interactions as expressed in *Bacillus subtilis* [67,68].

Further work is needed to understand why most of these prophages are less amenable to virulence gene carriage and any potential impact this may have on the evolution, adaptation and success of their host bacteria. For example, in *E. coli* genomes, the carriage of the Shiga toxin gene by the prophages can lead to lysogenic conversion of non-toxigenic strains to toxigenic ones [69,70]. Interestingly, lysogenic conversion has been reported in *K. pneumoniae*, and this is indeed critical to the bacterial pathogenicity [58]. Moreover, it is worthy of note that the presence of virulence genes in prophages, as seen in our study, does not necessarily translate to expression or lysogenic conversion in the strains [58]. This study provides strong evidence and supports foundational knowledge of the presence of virulence genes in the Kenyan prophages. However, more work is needed to determine if these prophages are inducible and capable of active infection and whether the genes are expressed in *K. pneumoniae.*

Our observation of the highest prophage carriage of eight per genome examined concurs with previous data that the majority of *K. pneumoniae* strains harbour 1–12 prophages in a single genome [30]. The average prophage carriage of 4.09 per genome that we observed also concurs with other findings [71] and generally placed the *K. pneumoniae* strains in the ‘low’ prophage carriage category, with up to ~5 prophages per genome using the prophage abundance classification system reported in *Salmonella* [23]. This is based on the average prophage counts in all genomes examined here, although this reflects differently if individual genomes were considered. Also, the observation that the prophage abundance was incomparable with the cumulative genome contribution per strain was reported previously in *K. pneumoniae* and *Bacillus thuringiensis* [30].

The overwhelming number of prophages observed and the strategy of analysing the genomes manually have restricted our work. It also restricted the number of prophages we analysed for virulence genes identification; thus, we focused on the intact prophages only due to their potential to infect bacteria, as shown previously [23]. However, we appreciate that this is extremely limiting and thus, work is currently ongoing to automate our methods, which would help to minimise human error, enhance our analysis and study significantly more bacterial genomes and explore both the questionable and incomplete prophages as well.

The phylogenetic clustering of the intact prophages into three main clades has been reported, and the interaction spanned across prophages that were extracted from *K. pneumoniae* strains identified from all the sources and locations as well [59]. This shows the plasticity of the phage genomes and their interactions with genes from diverse clinical and environmental sources.

## 4. Materials and Methods

### 4.1. Collation of Bacterial Genomes

The publicly available genomes of the *K. pneumoniae* strains (89 genomes) examined in this study were downloaded from the National Center for Biotechnology Information (NCBI) GenBank database under BioProject PRJNA77784 (https://www.ncbi.nlm.nih.gov/bioproject/?term=PRJNA777842 (accessed on 1 August 2024)) [19]. The *K. pneumoniae* genomes were part of a previously published AMR surveillance and environmental studies in Kenya that included isolates sourced from patients (with community-acquired and healthcare-associated infections) and from the environment of eight hospitals in five counties in Kenya [19].

### 4.2. Identification and Extraction of Prophages from the Kenyan K. pneumoniae Genomes

The first stage of the study here was to identify and extract the genomes of the prophages that are encoded in the 89 *K. pneumoniae* strains above. To accomplish this, the genomes were analysed using the PHAge Search Tool with Enhanced Sequence Translation tool (PHASTEST version 3.0, https://phastest.ca/) [72]. PHASTEST is a web-based phage annotation server that uses Prodigal software for the initial open reading frame (ORF) identification and the protein translation of prophage proteins and genomes in their bacterial hosts. This strategy offers an added advantage of much lower detection of false prophage-positive outputs compared with the previous versions of the bioinformatic tool, PHAST and PHASTER that use GLIMMER [72,73,74].

To extract the prophages from each bacterial strain, the individual genomes (nucleotide FASTA files) were uploaded to the PHASTEST server, and the options ‘Select bacterial sequence annotation mode: Lite’ and ‘My input consists of multiple separate contigs (FASTA format only)’ were checked before submissions. All outputs were saved according to the total number, completeness (intact, score > 90; questionable, score 70–90; and incomplete, score < 70) and contributory genomes capacities of the prophages present in each of the bacterial genomes from all the sources and locations. To ensure the elimination of human error, the PHASTEST analysis was conducted independently by two of the authors and the outputs compared. Where there were discrepancies, bacterial genomes were re-submitted to the server, and the analysis was repeated to ascertain accurate outputs. The data was analysed using GraphPad Prism 8. Afterwards, the intact prophages (which have the potential to infect bacteria) were targeted, and their corresponding genomes were downloaded and saved for further analyses, as demonstrated in a previous study [23].

### 4.3. Identification of Virulence Genes Within the Intact Prophages and Their Association with K. pneumoniae Pathogenicity

The intact prophage genomes were each examined for the presence of genes that code for virulence factors associated with *Klebsiella* via the VFDB of pathogenic bacteria https://www.mgc.ac.cn/cgi-bin/VFs/genus.cgi?Genus=Klebsiella (accessed on 20 September 2024). Briefly, the genomes of the extracted intact prophages were submitted to the BLAST search tool within the JavaScript-rich interface (https://www.mgc.ac.cn/cgi-bin/VFs/v5/main.cgi (accessed on 20 September 2024)) of the VFDB. The options ‘nucleotide sequences from VFDB full dataset (setB)’ and the ‘blastn- nucleotide query vs. nucleotide db’ program were selected to capture all possible virulence genes (beyond *K. pneumoniae*) as previously described [23]. Copies of various genes identified (with default thresholds of e-value: ≤ 1e-100 or lower; sequence identity: ≥80%; coverage: ≥80%) were collated together and linked to *Klebsiella* virulence factors (secretion system, adherence, regulation, iron uptake, antiphagocytosis, autotransport, immune modulation, invasion, toxin and cell surface) available within the database. The *K. pneumoniae* NTUH-K2044 (NC_012731_ chromosome and plasmid pK2044) were used as a reference (default) in VFDB. The data were analysed using GraphPad Prism 8.

### 4.4. Phylogenetic Relationships of Intact Prophages of K. pneumoniae

Multiple alignment of the nucleotide sequences of the intact prophages was conducted in Geneious Prime 2025.0, a bioinformatics desktop software for sequence data analysis and editing (https://www.geneious.com). In Geneious, the alignment was conducted using the ‘global alignment option with free end gaps’ at 65% similarity cost matrix and with linear topology. The phylogenetic tree was constructed using the neighbour-joining tree method, with a bootstrap replication of 500 in Geneious software. Finally, the visualisation and annotation of the phylogenetic tree were conducted in iTOL version 7.2.1, a web-based tree-editing program [75].

## 5. Conclusions

This study seeks to determine the diverse prophages of *K. pneumoniae* and the potential roles they play in the pathogenicity of the bacterium to complement control strategies. Due to the paucity of information in this area on isolates from SSA, here, 89 publicly available clinical and environmental *K. pneumoniae* genomes were investigated. The genomes were generated from a previous surveillance study involving eight hospitals in five counties in Kenya between the years 2015 and 2020. Genome analyses revealed that diverse prophages are found in 88 of the genomes, and a single strain lacked prophages in its genome. The prophage load averaged 4.09 in number and 2.8% contributory genome content in each prophage-encoding bacterial strain investigated. Intact prophages were more abundant than incomplete and questionable ones put together. Due to their potential to undergo infection and mediate horizontal gene transfer, the intact prophages were targeted and further investigated for virulence genes carriage. Only 30% of the intact prophages were found to encode genes that are linked to ten virulence factors (iron uptake, regulation, adherence, secretion system, antiphagocytosis, autotransporter, immune modulation, invasion, toxin and cell surface/capsule) of *K. pneumoniae.* Phylogenetic analysis of the intact prophages revealed three distinct clades. The prophages were distributed across the clades irrespective of the sources and locations of their bacterial hosts. This study provides novel insights into the potential role prophages play as mediators of virulence determinants in *K. pneumoniae* from this part of the world, affecting the bacterial pathogenicity and success. Further work is needed to show if these prophages are inducible and capable of lysogenic conversion in *K. pneumoniae.*

## Figures and Tables

**Figure 1 antibiotics-14-01145-f001:**
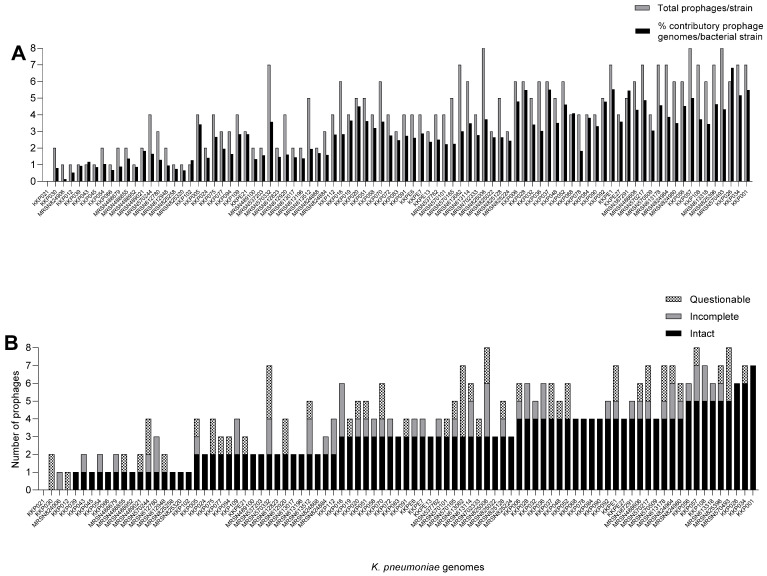
Prophage carriage in the 89 *K. pneumoniae* genomes from Kenya. (**A**) shows cumulative contributory genome contents of the prophages identified in each bacterial genome and (**B**) shows the number of complete, incomplete and questionable prophages found in each bacterial genome. The prophages were extracted using PHASTEST and data were analysed using GraphPad Prism 8.

**Figure 2 antibiotics-14-01145-f002:**
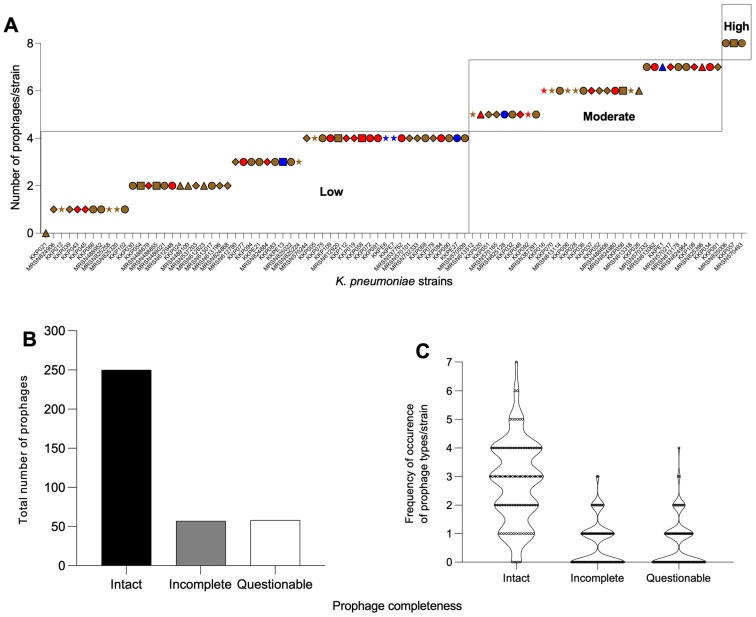
Diverse prophages found in the Kenyan *K. pneumoniae* genomes examined in this study. (**A**) Classification (based on a previous system [23]) of prophage abundance in each bacterial host genome from different sources and counties. Low = up to 5 prophages; moderate = 6–7 prophages; and high = 8–9 prophages encoded in the genomes of isolates from soft skin (brown), UTI (red) and hospital environment (blue) from the five counties: Kisumu (circle), Kilifi (triangle), Nairobi (diamond), Kisii (star) and Kericho (square). (**B**) Total number of intact (black), incomplete (grey) and questionable (white) prophages extracted from the genomes. (**C**) The frequency of occurrence of prophage types in all the genomes examined based on completeness. The prophages were extracted using PHASTEST and the data were analysed using GraphPad Prism 8.

**Figure 3 antibiotics-14-01145-f003:**
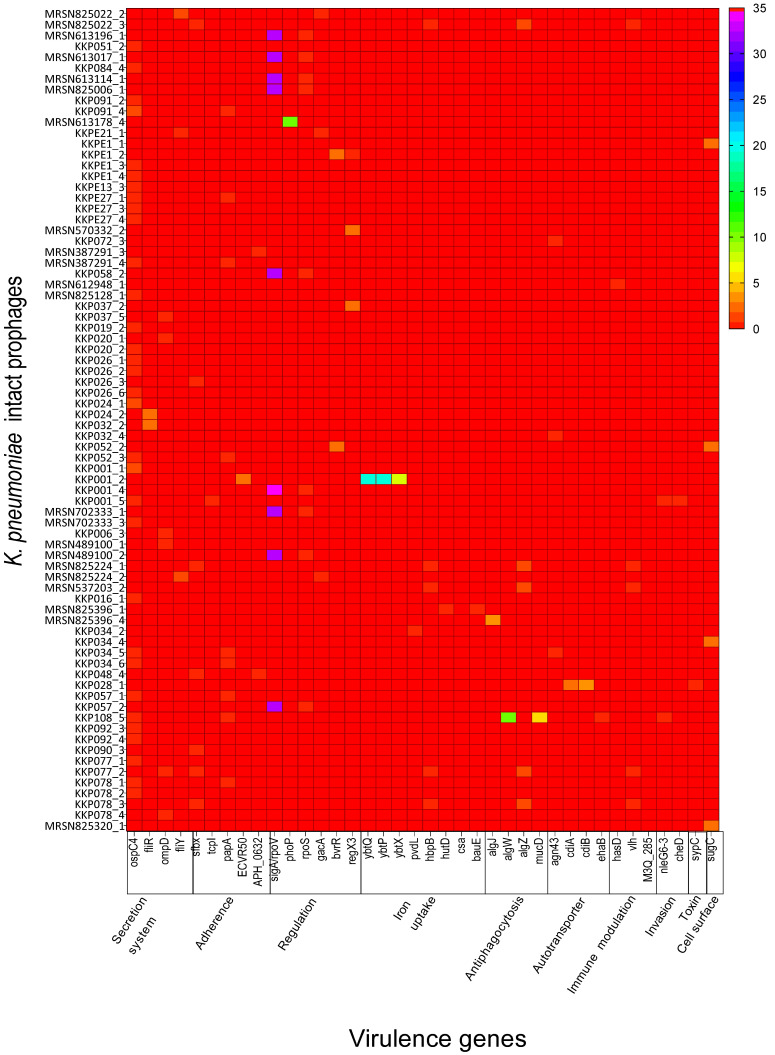
The 38 virulence genes identified in the genomes of the intact prophages of *K. pneumoniae* examined in this study are shown. Genes were identified using VFDB and classified based on functions and linked to 10 virulence categories and *K. pneumoniae* pathogenicity. The data were analysed using GraphPad Prism 8.

**Figure 4 antibiotics-14-01145-f004:**
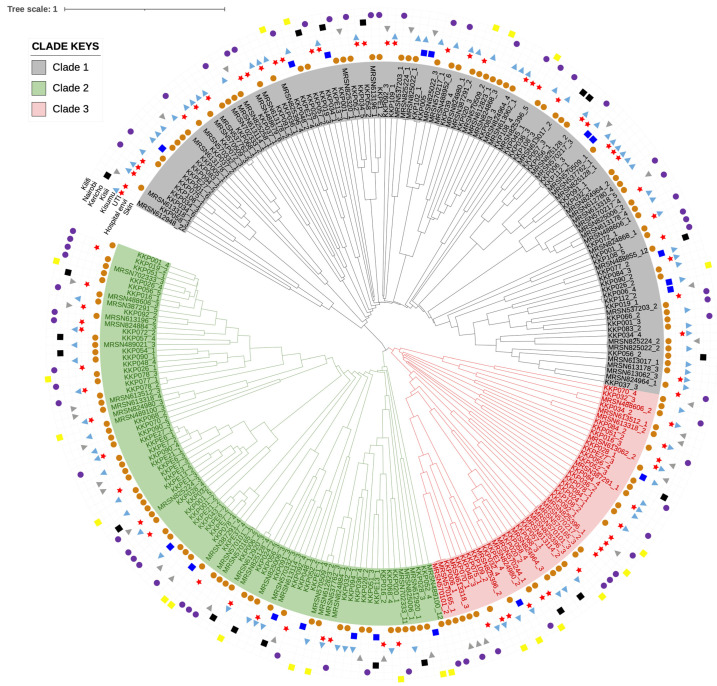
Phylogenetic tree showing the relationships of the 250 intact prophages extracted from the *K. pneumoniae* strains examined in this study. The three clades (Clade 1—grey; Clade 2—green; and Clade 3—pink) identified are shown. The sources and locations of the isolates from which the prophages were extracted are also shown. Prophage genomes were extracted using PHASTEST. Genome alignment was performed, and the tree was constructed using the neighbour-joining tree method with a bootstrap replication of 500 in Geneious Prime 2025.0. The tree was edited using iTOL 7.2.1, a web-based tree-editing program.

## Data Availability

Data are contained within the article and Appendix A.

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
