# Peer review of "The Potential Roles of Prophages in the Pathogenicity of Klebsiella pneumoniae Strains from Kenya"

_antibiotics, 2025, doi:10.3390/antibiotics14111145_

Round 1

Reviewer 1 Report

Comments and Suggestions for Authors

Identifying key genetic factors is imperative to combat antimicrobial resistance (AMR)  and its spread. In this manuscript the authors investigated the role of prophages in the pathogenicity of K. pneumoniae isolates from Kenya.  While this is an interesting study and can enhance the understanding of the role of the prophages in AMR, I have minor concerns regarding the reporting of data (detailed below):
1. The current writing has grammatical errors. Therefore, a professional round of language editing before the paper is published is highly recommended. The manuscript has room for improvement in terms of writing for clarity and smoother readibilty. Repetitions should be avoided. For eg., in line #425 and 427.
2. Line#28 - this study provides the first evidence
3. line #38 - "biofilm forming pathogen or bacterium" instead of biofilm former
4. line #41 add a list of antibiotics including cephalosporins, aminoglycosides and carbapenems
5. In conclusion authors should add some key findings of the study to strengthen it.
6. In results section "error reference not found" issue may have occured while converting the doc to PDF and should be corrected.

Author Response

Identifying key genetic factors is imperative to combat antimicrobial resistance (AMR)  and its spread. In this manuscript the authors investigated the role of prophages in the pathogenicity of K. pneumoniae isolates from Kenya.  While this is an interesting study and can enhance the understanding of the role of the prophages in AMR, I have minor concerns regarding the reporting of data (detailed below):
1. The current writing has grammatical errors. Therefore, a professional round of language editing before the paper is published is highly recommended.

Thank you. We have extensively reviewed the manuscript against grammatical errors. Please see updated manuscript.

 The manuscript has room for improvement in terms of writing for clarity and smoother readibilty. Repetitions should be avoided. For eg., in line #425 and 427.

Thank you for your observation. We re-wrote the conclusion section completely. For the particular sentence in question, the repetitions were corrected to reads ‘This study provides novel insights into the potential role prophages play as mediators of virulence determinants in K. pneumoniae from this part of the world, affecting the bacterial pathogenicity and success. Further work is needed to show if these prophages are inducible and capable of lysogenic conversion in the K. pneumoniae.' Lines 444-449 . We also searched the manuscript against more repetitions, and these were corrected as well.

  1. Line#28 - this study provides the first evidence

Thank you. This has been corrected. Please line 28-29 of the updated manuscript.

  1. line #38 - "biofilm forming pathogen or bacterium" instead of biofilm former

Thank you. This has been edited. Please line 38

  1. line #41 add a list of antibiotics including cephalosporins, aminoglycosides and carbapenems

Thank you. This has been corrected. Please see line 41-42, with an additional reference to support the extra information.

  1. In conclusion authors should add some key findings of the study to strengthen it.

The conclusion was re-written to reflect the key findings of the work. Please see lines 429-449

In results section "error reference not found" issue may have occurred while converting the doc to PDF and should be corrected.

Yes, these errors were generated due to the pdf conversion of the word-document. We have corrected these errors throughout the manuscript.

Thank you

Reviewer 2 Report

Comments and Suggestions for Authors

The manuscript addresses an important and underexplored topic: the contribution of prophages to the virulence and genetic diversity of Klebsiella pneumoniae isolates from Kenya. The study is timely and relevant given the global rise of antimicrobial resistance (AMR) and the limited data available from sub-Saharan Africa. The use of publicly available genomes and bioinformatic tools (PHASTEST, VFDB, Geneious) is appropriate. However, the manuscript would benefit from substantial improvement in structure, data presentation, and interpretation.

Point by point comments

Criteria used by PHASTEST to classify prophages as “intact,” “questionable,” or “incomplete” are not described. Maybe the authors may wish to validate prophage annotation with a secondary tool such as VirSorter2 or Prophage Hunter to confirm robustness.

The cut-off values (e.g., sequence identity, coverage, e-value thresholds in VFDB BLAST) are not specified.

No indication of genome quality control or assembly metrics (e.g., N50, contig number) is provided.

Clarify whether the prophage analysis considered plasmid sequences, since K. pneumoniae plasmids often carry phage elements.

Figure references (“Error! Reference source not found”) appear throughout, indicating missing or unlinked figures—this must be corrected before review.

The finding that 30% of intact prophages encode virulence genes is interesting, but the biological meaning remains speculative. The authors should:

  • Provide examples of prophages carrying specific virulence loci (with accession IDs or genomic coordinates).
  • Clarify whether any of these virulence genes are phage-encoded homologs or simply bacterial genes within prophage boundaries.
  • Discuss whether any of the virulence genes identified are known to be functional or experimentally validated in

The Discussion is sometimes repetitive and lacks critical analysis of limitations. It should:

  • Discuss possible biases from using only publicly available genomes, which may not represent all Kenyan isolates.
  • Compare prophage diversity with other Enterobacteriaceae (e.g., coli, Salmonella).
  • Explain the potential implications of prophage-mediated horizontal gene transfer for AMR dissemination.
  • The authors should moderate strong statements (e.g., “This study shows the role prophages play…”) because expression or function was not experimentally validated.

Ensure consistent italicization of species names (Klebsiella pneumoniae, Escherichia coli).

Author Response

Point by point comments

Criteria used by PHASTEST to classify prophages as “intact,” “questionable,” or “incomplete” are not described.

Thank you for the observation. We have added this in the methodology, lines 393-394

Maybe the authors may wish to validate prophage annotation with a secondary tool such as VirSorter2 or Prophage Hunter to confirm robustness.

Thank you for your suggestion of the tools above. We are aware of these tools, and we conducted a pilot study and found that PHASTEST is more amenable to our aims and objectives. We also tried PhageBoost. PHASTEST is able to classify the outputs into intact, incomplete or questionable which these tools con not provide. Our aim was to identify intact prophages which have the potential to infect, and to examined them for virulence genes carriage.

The cut-off values (e.g., sequence identity, coverage, e-value thresholds in VFDB BLAST) are not specified.

E-value: ≤ 1e-100 or lower, Sequence identity: ≥ 80%, Coverage: ≥ 80%. 

We apologise for this oversight. Please see above and we have have added these details to the methodology. Lines 412-413

No indication of genome quality control or assembly metrics (e.g., N50, contig number) is provided.

These were already conducted in the original publication of the genomes. Please see reference 19: 

(Muraya, A.; Kyany'a, C.; Kiyaga, S.; Smith, H.J.; Kibet, C.; Martin, M.J.; Kimani, J.; Musila, L. Antimicrobial

Resistance and Virulence Characteristics of Klebsiella pneumoniae Isolates in Kenya by Whole-Genome Sequencing. Pathogens 2022, 11, doi:10.3390/pathogens11050545)

Clarify whether the prophage analysis considered plasmid sequences, since K. pneumoniae plasmids often carry phage elements.

Thank you for your observation. Our focus was on prophages only.

Figure references (“Error! Reference source not found”) appear throughout, indicating missing or unlinked figures—this must be corrected before review.

Thank you. These occurred during PDF conversion of word document. We have corrected this error throughout our manuscript

The finding that 30% of intact prophages encode virulence genes is interesting, but the biological meaning remains speculative. The authors should:

  • Provide examples of prophages carrying specific virulence loci (with accession IDs or genomic coordinates).

Thank you for your observation. We provide examples of prophages of Klebsiella pneumoniae which encode some of the genes. See lines 325-330.

  • Clarify whether any of these virulence genes are phage-encoded homologs or simply bacterial genes within prophage boundaries.

These are phage-encoded homologs. Only the extracted prophage genomes are used for the virulence genes analyses. We agree that sometimes bacterial gene fragments are found in phages. See lines 389-401

  • Discuss whether any of the virulence genes identified are known to be functional or experimentally validated in

All the papers we cited for supported the carriage of some of these genes Klebsiella pneumoniae but there are paucity of information on their expression. We provide information on expression where available in other species eg Lines 335-342, 355. And we agree that the genes functions need to be ascertained, and we mentioned this in the manuscript. Please see lines 340-346

The Discussion is sometimes repetitive and lacks critical analysis of limitations. It should:

  • Discuss possible biases from using only publicly available genomes, which may not represent all Kenyan isolates.

We used the only publicly available genomes, which were isolated from a surveillance study and derived from different counties, sources and years. These genomes are excellent for comparisons from several sources and counties within the country. See lines 89-95, 372-378. We appreciate the limitation of our study due to the manual steps involved and we are working hard to automate these analyses which would enable us to examine more isolates.  See lines 357-364

  • Compare prophage diversity with other Enterobacteriaceae (e.g., coli, Salmonella).

Please see lines 270, 292-299, 335-340, 

  • Explain the potential implications of prophage-mediated horizontal gene transfer for AMR dissemination.

We focused on only virulence genes in this study. We appreciate this suggestion, but this is outside the context of our current study.

  • The authors should moderate strong statements (e.g., “This study shows the role prophages play…”) because expression or function was not experimentally validated.

We have moderated such statements throughout the manuscript. Pertinent to the phrase above we have re-written our conclusion. Lines 445-449

Ensure consistent italicization of species names (Klebsiella pneumoniae, Escherichia coli).

Thank you and we are sorry for these errors. These have been corrected throughout the manuscript. These errors came from the references, so we have corrected our library.  Please see reference section.  

Reviewer 3 Report

Comments and Suggestions for Authors

In their manuscript Khayeli et al. present a descriptive analysis of prophage presence and gene content in Klebsiella pneumonia (K.p.) strains from Kenia. The reader is informed on the number of prophages in the genomes of K.p. strains and virulence genes present in prophage sequences which is an important information, however a number of immediate questions of reviewers interest remain untouched such as follows:

-which AMR genes are present in K.p. strains and how distant are these genes located from prophage sequences in the genome, thus at what likelihood AMR genes could be transduced by induced prophages?

-how prophages are related to prophages present in other species?

-what are susceptible target strains of induced prophages and what are the expected integration frequencies and genomic integration sites?

-which stresses may induce prophages at which frequencies?

-what other mobile elements (e.g. plasmids) are present in K.p. strains and what ist the relative impact of prophages in relation to such other mobile elements with regard to strain drift in general and AMR spread in particular ?

Additional comments are:

-throughout the text error notes are displayed instead of references

-in Fig. 3. scripts are too small and only identified genes should be displayed without red squares, in Fig. 4 color symbols should be explained in the legend.

In the absence of new experimental results the authors should provide a much more thorough  insight and discussion of retrieved genomic data regarding the raised questions above.

Author Response

which AMR genes are present in K.p. strains and how distant are these genes located from prophage sequences in the genome, thus at what likelihood AMR genes could be transduced by induced prophages?

Thank you for your comments. This work focused on virulence genes carriage in in intact prophages. We appreciate that identification of AMR genes in the bacterial genomes in relation to the location of prophages would provide additional information, but this is outside the scope and context of our current work.

-how prophages are related to prophages present in other species?

Please see examples of prophage carriage in other species: C. difficile -Lines 270, Bacillus subtilis-lines 334, Bacillus thuringiensis-line- 356, Salmonella-lines 292-299, 352 E. coli-lines 270, 335-339

-what are susceptible target strains of induced prophages and what are the expected integration frequencies and genomic integration sites?

The integration sites and frequencies are variable across species and strains. Although our, we did not focus on integration sites but we provided information (273-277) in the discussion as advised. .

-which stresses may induce prophages at which frequencies?

Thank you for your comment. We provide comments on prophage induction. See lines 305-310

-what other mobile elements (e.g. plasmids) are present in K.p. strains and what ist the relative impact of prophages in relation to such other mobile elements with regard to strain drift in general and AMR spread in particular ?

Thank you for your comment. Our work focused on prophages alone and their role in K. pneumoniae pathogenicity. We understand that plasmids can mediate gene transfer, but this is outside the context of study here. 

Additional comments are:

-throughout the text error notes are displayed instead of references.

These were errors relating to figure citations and we have corrected these throughout the manuscript.

-in Fig. 3. scripts are too small and only identified genes should be displayed without red squares,

Thank you. We have removed all prophages without any virulence genes and edited the graph for the scripts to be legible.  Please see updated figure 3

in Fig. 4 color symbols should be explained in the legend.

Thank you. These have been added to lines 251 for figure 4 and lines 154-157 for figure 2.

In the absence of new experimental results the authors should provide a much more thorough insight and discussion of retrieved genomic data regarding the raised questions above.

Thank you for your suggestion. Unfortunately, we do not have additional data on AMR carriage and plasmids as these are outside the scope of this work. We appreciate these are good suggestions and we would consider them in the future.  Hence, we elaborated more one

  1. integration frequencies and genomic integration sites lines (lines 273-277)
  2. prophage carriage in other species. C. difficile -Lines 270, Bacillus subtilis-lines 334, Bacillus thuringiensis-line- 356, Salmonella-lines 292-299, 352 E. coli-lines 270, 335-339
  3. stresses induction of prophages and frequencies (305-310)

Thank you.

Round 2

Reviewer 2 Report

Comments and Suggestions for Authors

It has been well revised.

Reviewer 3 Report

Comments and Suggestions for Authors

The raised points have been briefly addressed.